# Impaired eIF5A function causes a Mendelian disorder that is partially rescued in model systems by spermidine

Víctor Faundes [1,2], Martin D. Jennings [3,4], Siobhan Crilly [5], Sarah Legraie[1], Sarah E. Withers[5], Sara Cuvertino[1], Sally J. Davies[6], Andrew G. L. Douglas[7,8], Andrew E. Fry[6,9], Victoria Harrison[7], Jeanne Amiel[10,11,12], Daphné Lehalle[10], William G. Newman [1,13], Patricia Newkirk[14], Judith Ranells[14], Miranda Splitt [15], Laura A. Cross [16,17], Carol J. Saunders[18,19,20], Bonnie R. Sullivan [16,17], Jorge L. Granadillo[21], Christopher T. Gordon [11,12], Paul R. Kasher[4,5✉], Graham D. Pavitt [3,4✉] & Siddharth Banka [1,13✉]

The structure of proline prevents it from adopting an optimal position for rapid protein synthesis. Poly-proline-tract (PPT) associated ribosomal stalling is resolved by highly conserved eIF5A, the only protein to contain the amino acid hypusine. We show that de novo heterozygous *EIF5A* variants cause a disorder characterized by variable combinations of developmental delay, microcephaly, micrognathia and dysmorphism. Yeast growth assays, polysome profiling, total/hypusinated eIF5A levels and PPT-reporters studies reveal that the variants impair eIF5A function, reduce eIF5A-ribosome interactions and impair the synthesis of PPT-containing proteins. Supplementation with 1 mM spermidine partially corrects the yeast growth defects, improves the polysome profiles and restores expression of PPT reporters. In zebrafish, knockdown *eif5a* partly recapitulates the human phenotype that can be rescued with 1 μM spermidine supplementation. In summary, we uncover the role of eIF5A in human development and disease, demonstrate the mechanistic complexity of *EIF5A*-related disorder and raise possibilities for its treatment.

A list of author affiliations appears at the end of the paper.

Proline is a unique amino acid as its amine nitrogen is bonded to two, instead of one, carbon atoms with a distinctive rigid cyclic structure that prevents it from adopting an optimal position required for rapid protein synthesis[1]. The presence of proline, either as a peptidyl donor or an acceptor, impedes the rate of peptide bond formation by the ribosome, an inhibitory effect that becomes progressively stronger, to the extent that three or more consecutive prolines provoke ribosome stalling[2–4]. In eukaryotic cells, ribosomal stalling is resolved by the Eukaryotic Translation Initiation Factor 5A (eIF5A), which is critical for the synthesis of peptide bonds between consecutive proline residues[5]. Notably, the frequency of poly-proline tracts (PPTs) is higher in evolutionarily new proteins and of all tandem amino acid repeats, only the proline repeat frequency correlates with functional complexity of eukaryotic organisms[6]. eIF5A1 (hereafter eIF5A) and its normally undetectable paralogue eIF5A2 are the only human proteins that contain the amino acid hypusine, a post-translationally modified lysine at position 50 (K50)[2–4]. Hypusine is synthesised from spermidine, a polyamine, via two sequential enzymatic steps involving highly conserved deoxyhypusine synthase (DHPS) and deoxyhypusine hydroxylase (DOHH) enzymes[7]. Hypusinated eIF5A stabilises P-tRNA that facilitates peptide bond formation at stalled ribosomes[5]. Other functions of eIF5A include recognition of the correct start codon[8,9], global protein synthesis elongation and termination[10,11], promoting the elongation of many non-poly-proline-specific tripeptide sequences, and eliciting nonsense-mediated decay (NMD)[12]. It is essential for cell viability and growth in both simple[2,5] and complex[3] organisms. Somatic overexpression of EIF5A has unfavourable prognostic implications in several cancers, including pancreatic, lung, hepatocellular, bladder, and colorectal carcinomas[13,14]. However, no human phenotype has been previously attributed to germline EIF5A variants.

Here, we demonstrate that de novo heterozygous EIF5A variants cause a previously undescribed syndrome characterised by variable combinations of developmental delay, microcephaly, micrognathia, congenital malformations and dysmorphism. These variants likely result in the loss of eIF5A function through distinct mechanisms and lead to impaired interaction between eIF5A and ribosomes. Both functional and phenotypic consequences of impaired eIF5A activity are partially rescued by spermidine supplementation in yeast and zebrafish models.

## Results

**Germline variants in EIF5A cause a previously undescribed craniofacial-neurodevelopmental disorder.** In an individual with intellectual disability, congenital microcephaly, micrognathia and clinical suspicion of a Kabuki syndrome (MIM # 147920)-like condition[15], we identified a de novo heterozygous frameshift variant in EIF5A (c.324dupA, p.R109Tfs*8) by trio whole-exome sequencing (Fig. 1a–c, Table 1, individual #3, and Supplementary Note 1). EIF5A has four protein-coding transcripts, but ENST00000336458/NM_001970.5 (Uniprot P63241) is preferentially, widely and most highly expressed in all adult human tissues[16–18]. Only one truncating variant has been recorded in gnomAD, a variant sequence database of >140,000 control individuals, in this transcript (gnomAD $o/e_{LoF} = 0.11$; pLI = 0.74)[19]. Also, the GeVIR metrics (GeVIR AD = 4.38; VIRLoF AD = 2.21) add supportive evidence that EIF5A is highly likely to be associated with autosomal dominant disease[20]. Hence, we concluded that the EIF5A variant identified in individual #3 was significant. Through Matchmaker Exchange[21] and GeneMatcher[22] we identified six additional individuals with nonsense or missense de novo EIF5A variants (c.143C>A, p.T48N; c.316G>A, p.G106R; c.325C>G, p.R109G; c.325C>T, pR109*; c.343C>T, p.P115S; c.364G>A, p.E122K). The missense variants are

located in one of the most constrained coding regions of the human genome (>99th percentile)[23], and affect residues that are highly evolutionarily conserved (Fig. 1b, c). In silico modelling of missense variants (Fig. 1d) onto the structure of yeast eIF5A in complex with the 60S ribosome (PDB entry 5GAK)[24] shows that the missense variants affect surface-exposed residues. T48 is adjacent to the hypusinated lysine 50, G106 and R109 residues are close to the ribosomal protein uL1 and E122 is close to the P-site tRNA. In contrast, P115 has no clear intermolecular interactions. Although the individuals were identified via their genotypes, on reverse phenotyping[25] their clinical features showed remarkable convergence. All patients were affected by variable degrees of developmental delay and/or intellectual disability, microcephaly (either absolute or relative) and overlapping facial dysmorphisms (Table 1, Fig. 1a and Supplementary Note 1). Notably, four individuals in this cohort were clinically suspected to have either a Kabuki syndrome-like or a mandibulofacial dysostosis (MIM #154400)-like condition (Table 1 and Supplementary Note 1). eif5a/Eif5a mRNA is highly expressed in Danio rerio (hereafter zebrafish)[26] and Mus musculus[27] (hereafter mouse) embryos in structures that form the brain and mouth, thus corresponding with the most significantly impacted structures in the affected individuals. Overall, these results are strongly indicative of the frameshift, nonsense and missense EIF5A variants being causal for the phenotypes of the affected individuals.

**EIF5A variants impair eIF5A function.** Peripheral blood mononuclear cells obtained from Individual 3, with the EIF5A frameshift variant c.324dupA, and two healthy controls were transformed by Epstein-Barr virus into lymphoblastoid cell lines (LCLs). Blood samples from other affected individuals were not available. The EIF5A mRNA level in LCLs was significantly reduced in Individual 3 (Fig. 1e) and the transcript with c.324dupA was not detected (Supplementary Fig. 1), suggesting NMD of the mutant transcript.

The Saccharomyces cerevisiae (yeast) and human eIF5A share a very high degree of conservation (62% identity/92% similar) (Fig. 1b) and well-established assays to investigate eIF5A function in yeast are available[2,5]. We synthesised a human EIF5A cDNA (heIF5A hereafter) optimised for yeast codon usage, and added the 5′ and 3′ control regions of the yeast homologue of EIF5A (known as TIF51A or HYP2, yeIF5A hereafter) (Supplementary Note 2 and Supplementary Table 1). Introduction of this construct on a centromeric plasmid using standard techniques in a yeast strain in which both TIF51A and TIF51B (a second yeIF5A gene that is transcribed only in anaerobic conditions[2]) are deleted (Supplementary Table 3 and Supplementary Fig. 2) restored its growth potential similar to the wild-type yeIF5A (Fig. 2a, rows 1 and 3). This confirmed that the synthetic heIF5A can replace yeIF5A functions in line with previous reports[28].

Next, we performed site-directed mutagenesis to create heIF5A constructs with the p.T48N, p.G106R, p.R109Tfs*8 and p.E122K variants in centromeric plasmids (Supplementary Tables 1 and 2). The individuals with the p.R109G, p.R109* and p.P115S variants were identified after experiments for other variants were concluded and, therefore, these variants were not included in the functional studies. Yeast colonies expressing the heIF5A-R109Tfs*8 as the sole source of eIF5A could not be obtained after ten days of growth, and across multiple plasmid shuffling experiments (Supplementary Figs. 2 and 3). Western blotting (WB) using a monoclonal anti-human eIF5A and an anti-hypusinated eIF5A antibody (hereafter hypusine) of pre-shuffled heIF5A-R109Tfs*8 strains co-expressing heIF5A-WT revealed that p.R109Tfs*8 was poorly expressed, and even when expressed in high-copy it was very poorly hypusinated (Supplementary Fig. 4). The poor expression of this mutant in yeast

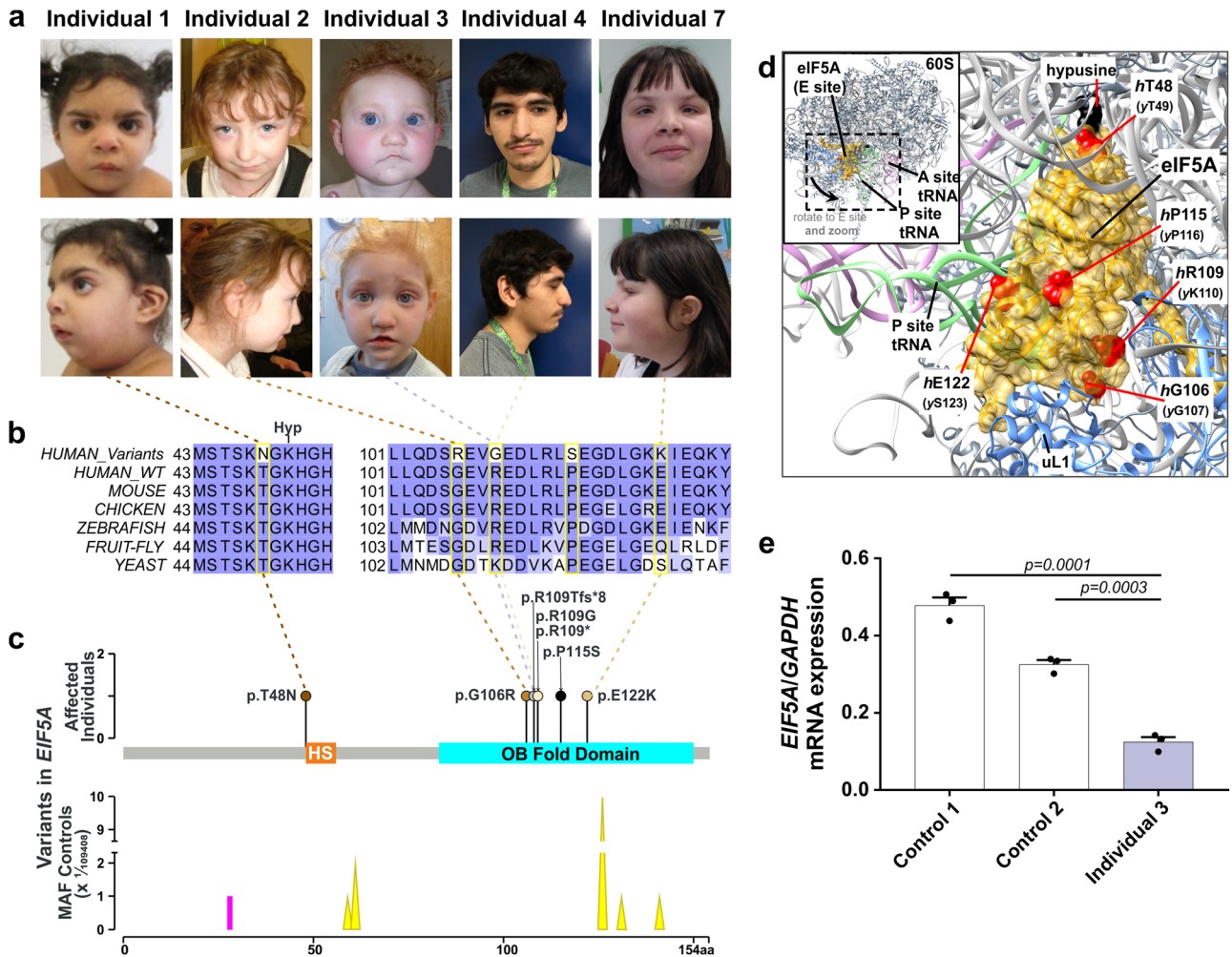

**Fig. 1 Heterozygous variants in *EIF5A* cause a novel craniofacial-neurodevelopmental disorder. a** Individuals with de novo heterozygous missense (individuals 1, 2, 4 and 7) or frameshift (individual 3) *EIF5A* variants display similar facial dysmorphism, microcephaly and micrognathia. Photographs of individuals 5 and 6 are not available. **b** The missense *EIF5A* variants affect highly conserved residues. Evolutionary conservation of residues affected by variants (delimited by yellow rectangles) is shown in five species. BLOSUM62 scores are depicted in purple scale (dark purple = completely conserved, light purple = relatively conserved, white = not conserved). Lysine (K) 50 is hypusinated (Hyp). **c** The *EIF5A* variants are novel and located in the functional sites and domains of eIF5A. Top: hypusine site (HS) is the orange bar between residues 48 and 55 (InterPro entry P63241), and the OB fold domain is the cyan bar between residues 83 and 150 (InterPro entry P63241). Bottom: the location and the minor allele frequencies (MAF) of high quality non-flagged missense variants (yellow triangles) and a protein-truncating variant (p.N28Mfs*64, magenta line) seen only in controls of gnomAD v2.1.1 for the transcript ENST00000336458. **d** In silico modelling of missense variants supports their deleterious nature. Variants are indicated as red spheres at the surface of yeast eIF5A (semi-transparent gold) with hypusinated K51 (human K50, *h*K50) (black spheres) shown bound to the yeast 60S ribosomal subunit E site (grey secondary structures only, with uL1 in blue), with adjacent P-site (green) and A-site (pink) tRNAs (PDB entry 5gak). Note that T48 (yeast T49, *y*T49) is in proximity to the hypusinated *h*K50; *h*G106 (yG107) and *h*R109 (yK110) are close to uL1 and *h*E122 (yS123) is close to the P-site tRNA. **e** The mutant *EIF5A* transcript with the truncating variant is not expressed. The mRNA levels in lymphoblastoid cells from individual 3 compared to control 1 (healthy female) and control 2 (healthy male) are shown relative to *GAPDH* ($2^{-\Delta CT}$ method). Each data point corresponds to one technical replicates, and the bars show the mean + SEM. Two-sided *P* values were determined by unpaired *t*-test.

is consistent with our inability to detect this protein in LCLs from individual 3. In contrast, the three missense mutations each supported yeast growth as the sole source of eIF5A showing that they retain sufficient eIF5A function for cell viability. The *h*eIF5A-T48N and *h*eIF5A-G106R yeast cells exhibited slow growth (Fig. 2a, rows 4 and 5), similar to a previously characterised temperature-sensitive mutant *y*eIF5A-S149P mutant (Fig. 2a, row 2)[2,5]. No significant difference was observed between growth of *h*eIF5A-E122K and *h*eIF5A-WT (Fig. 2a, rows 6). These results indicate that the p.T48N and p.G106R variants result in partial loss of eIF5A function, and p.R109Tfs*8 variant results in complete loss of viability. However, the impact of the p.E122K variant remained uncertain.

***EIF5A* variants reduce eIF5A–ribosome interaction through different mechanisms**. Next, we asked if the missense variants affected interaction of *h*eIF5A with ribosomes through polysome profiling (Supplementary Fig. 5). All missense variants that were tested, including the p.E122K variant, exhibited aberrant polysome profiles, with elevated polysome-to-monosome (P/M) ratios and an increase in the free 60S peak heights and the 60 S/80S ratios (Fig. 2c), consistent with a global translation elongation defect[2,5]. Probing for eIF5A and hypusine across polysome fractions revealed that WT eIF5A has ribosome-free (lanes 1–2 in Fig. 2c) and 80S (lane 6 in Fig. 2c) peaks. In contrast, each missense mutant showed a reduction in the 80S ribosome fraction (Fig. 2c), indicating a reduction in ribosome binding by each

**Table 1 Phenotypes of patients with EIF5A variants.**

| Characteristics | Individual | | | | | | |
|---|---|---|---|---|---|---|---|
| | 1 | 2 | 3 | 4 | 5 | 6 | 7 |
| Sex (age)[a] | F (6.9 y) | F (8.4 y) | F (8.4 y) | M (18.3 y) | M (8 mo) | M (4 y) | F (16.4 y) |
| Genomic position[b] | 17:7213097 | 17:7214714 | 17:7214722 | 17:7214723 | 17:7214723 | 17:7214741 | 17:7214762 |
| cDNA[c] protein consequence[d] | c.143C>A p.T48N | c.316G>A p.G106R | c.324dupA p.R109Tfs*8 | c.325C>G p.R109G | c.325C>T p.R109* | c.343C>T p.P115S | c.364G>A p.E122K |
| Inheritance/zygosity | DN Het | DN Het | DN Het | DN Het | DN Het | DN Het | DN Het |
| **Perinatal history** | | | | | | | |
| Congenital microcephaly | Yes | Yes | Yes | Unknown | No | No | Unknown |
| IUGR | Yes | Yes | No | Yes | No | No | No |
| Feeding difficulties | No | Yes | Yes | Yes | Yes | No | No |
| Other | No | Cardiac anomalies | Cardiac anomalies Cleft palate | Hypotonia | Cardiac anomalies Hypotonia | No | Foetal ascites |
| DD/ID | Moderate/severe | Moderate | Mild | Moderate | Moderate | Mild/Moderate | Moderate |
| CNS anomalies | No | No | No | Peritrigonal hyperintensities | No | Left lateral ventriculomegaly | No |
| **Physical anomalies** | | | | | | | |
| Heart | No | Yes | Yes | Unknown | Yes | Unknown | Unknown |
| Craniofacial | Yes | No | Yes | Yes | Yes | No | No |
| Other | Hemivertebrae (L3) | No | No | Cryptorchidism Pes planus | No | No | Toe contractures Small toenails Pes planus |
| **Growth parameters** | | | | | | | |
| Height (SD) | N (0.53) | SS (−2.82) | N (0.86) | N (−0.45) | N (−1.45) | N (+1 SD) | SS (−2.59) |
| Weight (SD) | OW (2) | LW (−2.28) | N (−0.41) | N (0.06) | LW (−3.14) | N (−0.09 SD) | N (−0.69) |
| HC (SD) | Mi (−3) | Mi (−7.47) | Mi (−2.11) | Mi (−2.62) | N (−0.45) | N (−1.09) | Mi (−1.94) |
| **Facial dysmorphisms** | | | | | | | |
| Broad eyebrows | Yes | Yes | Yes | Yes | No | Yes | No |
| Abn. supraorbital ridges | Yes | Yes | Yes | Yes | No | No | No |
| Epi/telecanthus | No | Yes | Yes | Yes | Yes | No | Yes |
| Bulbous nasal tip | Yes | Yes | Yes | Yes | No | No | No |
| Thin upper lip | No | Yes | Yes | Yes | Yes | No | No |
| Micrognathia | Yes | Yes | No | Yes | No | Yes | No |
| Low set ears | Yes | Yes | No | No | No | No | Yes |
| Other | Lower eyelid hypoplasia | Hypertelorism | No | Prominent long ears | Plagiocephaly Sparse scalp hair Frontal bossing Downslanting PF Cupped ears | Long PF hypoplastic ala nasi | Deep-set eyes Abn. lower eyelids Small ears |
| **Other medical issues** | | | | | | | |
| Joint hypermobility | No | No | Yes | Yes | N/A | No | Yes |
| Eye anomalies | Yes | No | No | Yes | No | Yes | Yes |
| Others | No | Constipation Gastroesophageal reflux Gastrostomy | Conductive deafness Premature thelarche | Constipation | Dysphagia Gastrostomy Failure to thrive | Hypotonia Flat feet | Autism ADHD Delayed puberty Nasal polyps |
| Initial clinical suspicion | Mandibulofacial dysostosis like | Kabuki syndrome like | Kabuki syndrome like | Kabuki syndrome like | Mowat Wilson like | None | None |

Abn abnormal, ADHD attention deficit hyperactivity disorder, OW overweight, CNS central nervous system, DD developmental delay, HC head circumference, Het heterozygous, ID intellectual disability, IUGR intra-uterine growth retardation, LW low weight, Mi microcephaly, N within normal ranges, N/A not applicable, PF palpebral fissures, SD standard deviation, SS short stature.
[a]At last examination.
[b]According to hg19.
[c]GenBank reference NM_001970.5, Ensembl reference ENST00000336458.
[d]UniProtKB reference P63241-1.

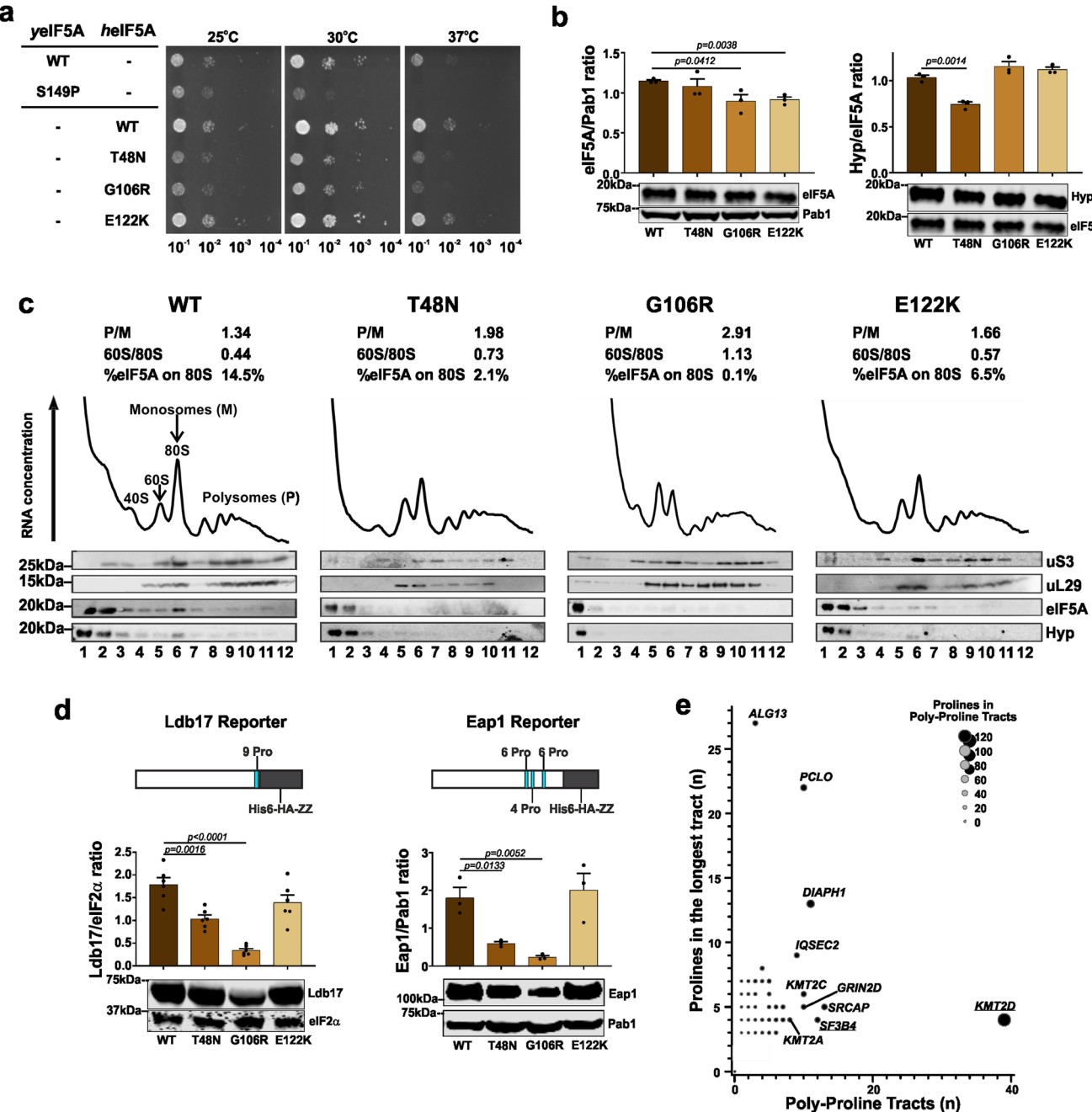

mutant eIF5A, consistent with the idea that impaired eIF5A–ribosome interactions impact translation elongation of each mutant.

To explore possible mechanisms for the reduction in ribosome binding of missense variants, we performed WB. In cells expressing *h*eIF5A-T48N, levels of total eIF5A were normal, but levels of hypusination were reduced, suggesting that the p.T48N impairs hypusination of the adjacent K50 residue. As hypusination is necessary for eIF5A function, reduced hypusination may contribute to the observed reduction in ribosome association. In contrast, total levels of eIF5A were modestly reduced in *h*eIF5A-G106R and *h*eIF5A-E122K cells, while hypusination was unaffected (Fig. 2b). Overall, these results indicated that *EIF5A* variants reduce eIF5A–ribosome interaction, likely through different mechanisms.

**EIF5A variants impair synthesis of proteins with PPTs.** Next, we examined if impaired eIF5A function impacted on translation of specific mRNAs. Because eIF5A is especially critical for the optimal synthesis of proteins containing PPTs[2,5], we studied expression of two previously described PPT reporters—a hae-magglutinin (HA)-tagged Ldb17 with a single long PPT of nine consecutive prolines and HA-Eap1 with three shorter PPTs (Supplementary Tables 1 and 3 and Fig. 2d)[5]. As these reporters require expression with galactose as a carbon source, we first evaluated whether levels of eIF5A and hypusination in these cell growth conditions were consistent with our previous findings. Here in both of the slow-growing mutants (p.T48N and p.G106R) expression of eIF5A was lower than WT. As previously, hypusination was consistently low for p.T48N (Supplementary Figs. 6 and 7). By WB of total cell extracts we observed decreased levels

**Fig. 2 EIF5A variants impair eIF5A function, its interaction with ribosome and synthesis of proteins with poly-proline tracts. a** Variants p.T48N and p.G106R affect yeast growth. Representative serial dilution growth assay of yeast strains for human eIF5A (*h*eIF5A) and its variants, compared to the growth of strains with WT yeast eIF5A (*ye*IF5A) or the thermosensitive *ye*IF5A-S149P after 2 days. Eighteen replicates per temperature were performed. **b** Variant p.T48N reduces eIF5A hypusination, whereas p.G106R and p.E122K decrease eIF5A levels. eIF5A expression (left) and hypusination (Hyp; right) among yeast strains with missense *h*eIF5A variants, grown at 30 °C in synthetic complete (SC) liquid medium. Each data point corresponds to one biological replicate, which was controlled as stated in the 'Methods' (see Eq. (1)), and the bars show the mean + SEM. Two-sided *P* values were determined by unpaired *t*-test. Full uncropped images of gel blots are shown in Supplementary Fig. 9. **c** Missense mutants decrease eIF5A interaction with ribosome. Polysome profiles of yeast expressing either *h*eIF5A-WT or missense variants, grown at 30 °C in SC medium. Corresponding western blot analyses of eIF5A, hypusine, and the ribosomal proteins uS3 and uL29 probed across gradient fractions for each polysome profile are presented beneath. Polysome-to-monosome (P/M) and 60/80S ratios for the $A_{260}$ traces are given, calculated by comparing the areas under the 80S and polysome peaks. In addition, the fraction of total eIF5A signal associated with the 80S (western blot lane 6) is given. Full uncropped images of gel blots are shown in Supplementary Fig. 10. One profile was performed. **d** Missense mutants decrease synthesis of reporters with poly-proline tracts. Comparison of Ldb17 (left graph) and Eap1 (right graph) poly-proline-containing reporter expression in *h*eIF5A yeast strains grown at 30 °C in SCGal medium. Data presentation and statistical treatment as described for panel **b**. Full uncropped images of gel blots are shown in Supplementary Fig. 11. **e** Genes with the highest numbers of prolines match the initial clinical suspicion for studied individuals. Enrichment of PPTs in MAGs. For each MAG the number of PPTs (*X*-axis) is plotted against the number of prolines in the longest PPT (*Y*-axis). MAG circle size represents the total number of prolines in PPTs in each protein. The 'top 10' ranked MAGs are named. Heterozygous loss-of-function variants in *SF3B4* and *KMT2D* (both underlined) cause acrofacial dysostosis 1, Nager type (a subtype of mandibulofacial dysostosis, MIM #154400) and Kabuki syndrome 1 (MIM #147920), respectively, which overlap with the initial clinical suspicions in individual 1, and for individuals 2 to 4, respectively.

of both PPT reporters for the p.T48N and p.G106R alleles (Fig. 2d). Although levels of Ldb17 in the p.E122K variant were typically lower than in WT cells, this did not reach statistical significance (Fig. 2d and Supplementary Fig. 7). Hence, the human eIF5A variants impair the synthesis of proteins containing PPTs in yeast.

**Human microcephaly-associated genes (MAGs) are enriched for PPTs.** Next, we explored if impaired synthesis of proteins with PPTs could help explain microcephaly, which was the most consistent feature of our patients. To study this, we prepared a catalogue of all known human MAGs according to OMIM and assessed their PPT content[29,30] (Supplementary Data 1). We observed that 198/685 (28.9%) MAGs and 4366/17981 (24.2%) of all other human protein-coding genes have ≥1 PPT [$\chi^2$ 7.64; OR = 1.27; 95% CI 1.07–1.5; *P* = 0.0057]). Next we ranked MAGs according to their proline content in PPTs (Supplementary Data 1 and Fig. 2e). *KMT2D* was ranked as #1 in this list. Loss-of-function *KMT2D* variants cause Kabuki syndrome[31], which was the clinically suspected diagnosis in three individuals. *SF3B4*, variants in which cause acrofacial dysostosis 1, Nager type[32] (phenotypic overlap with the clinical suspicion in individual 1), was ranked #5 (Table 1 and Fig. 2e).

**Spermidine partially rescues impaired eIF5A function in yeast.** Polyamines contribute to the efficiency and fidelity of protein synthesis, and spermidine may overcome absence of eIF5A to some extent to promote peptide synthesis[33]. Furthermore, DHPS mediated transfer of a 4-aminobutyl moiety from a polyamine, spermidine, to K50 is the first step in formation of active hypusinated eIF5A[34]. Previous work has demonstrated that spermidine promotes longevity in yeast, whereas depletion of polyamines has a deleterious effect[35]. We therefore reasoned that spermidine supplementation could potentially overcome the effects of impaired eIF5A function. In addition, because p.T48N hypusination was reduced, it may indicate that polyamine concentrations were limiting for hypusination in our growth conditions. We screened the effect of supplementing growth medium with different concentrations of spermidine on yeast growth, as measured by the rate of colony formation (Fig. 3a and Supplementary Fig. 8). One millimolar spermidine partially corrected the growth defects of p.T48N and p.G106R cells (Fig. 3a). As a growth phenotype was not observed in yeast expressing the

p.E122K variant, this assay was uninformative for this allele. Higher spermidine concentrations had progressively deleterious effects impairing the growth of all strains (Fig. 3a and Supplementary Fig. 8).

Next, we performed polysome profiling of WT, p.T48N and p.G106R cells in the presence of 1 mM spermidine. Spermidine treatment improved the global polysome profiles for both mutant strains with no impact on *h*eIF5A-WT (compare Fig. 3b with Fig. 2c). We observed full or partial restoration of *h*eIF5A interaction with the 80S ribosome in p.T48N and p.G106R cells, respectively (Fig. 3b). Furthermore, 1 mM spermidine restored expression of PPT reporter in both growth-rescued mutants (Fig. 3c, left graph). However, improved growth and protein synthesis was not explained by improved eIF5A expression levels or by enhanced hypusination of p.T48N mutant eIF5A (Fig. 3c, middle and right graph). These results suggest that spermidine can rescue and/or bypass impaired eIF5A functions in protein synthesis independent of its role as a substrate for hypusination of eIF5A K50 (ref. [33]).

**Spermidine partially rescues phenotypes of impaired eIF5A function in a zebrafish model.** We next investigated if spermidine can rescue the impact of loss of eIF5A function in a developing vertebrate model. Zebrafish *eif5a* shares a high degree of conservation with its human orthologue (74% identity, 86% similarity) (Fig. 1b). Previous studies have demonstrated that morpholino-mediated knockdown of *eif5a* or transient overexpression of human *EIF5A* can cause microcephaly in zebrafish larvae[36]. We used a validated and published splice site morpholino (MO)[36] to knockdown *eif5a* in fertilised nacre[37] zebrafish eggs, which were incubated in a standard, spermidine-free medium. The resulting larvae were fixed and cartilage was stained at 77 h post-fertilisation (hpf). In line with the human disorder phenotypes, we measured the distance between irises (translating to head circumference and therefore serving as a model for microcephaly) and the length of mandible cartilages (translating to mandibular growth and therefore serving as a model for micrognathia) (Fig. 4a, left and right photographs, respectively). Although we did not recapitulate the previously described microcephaly phenotype[36] (Fig. 4b, c, left graph), the *eif5a* MO induced micrognathia in zebrafish larvae (Fig. 4b, c, right graph). Next, we explored the effects on zebrafish of supplementation with 10-fold dilutions (from 1 mM to 0.1 μM) of spermidine. One hundred micromolar or higher concentrations killed larvae before

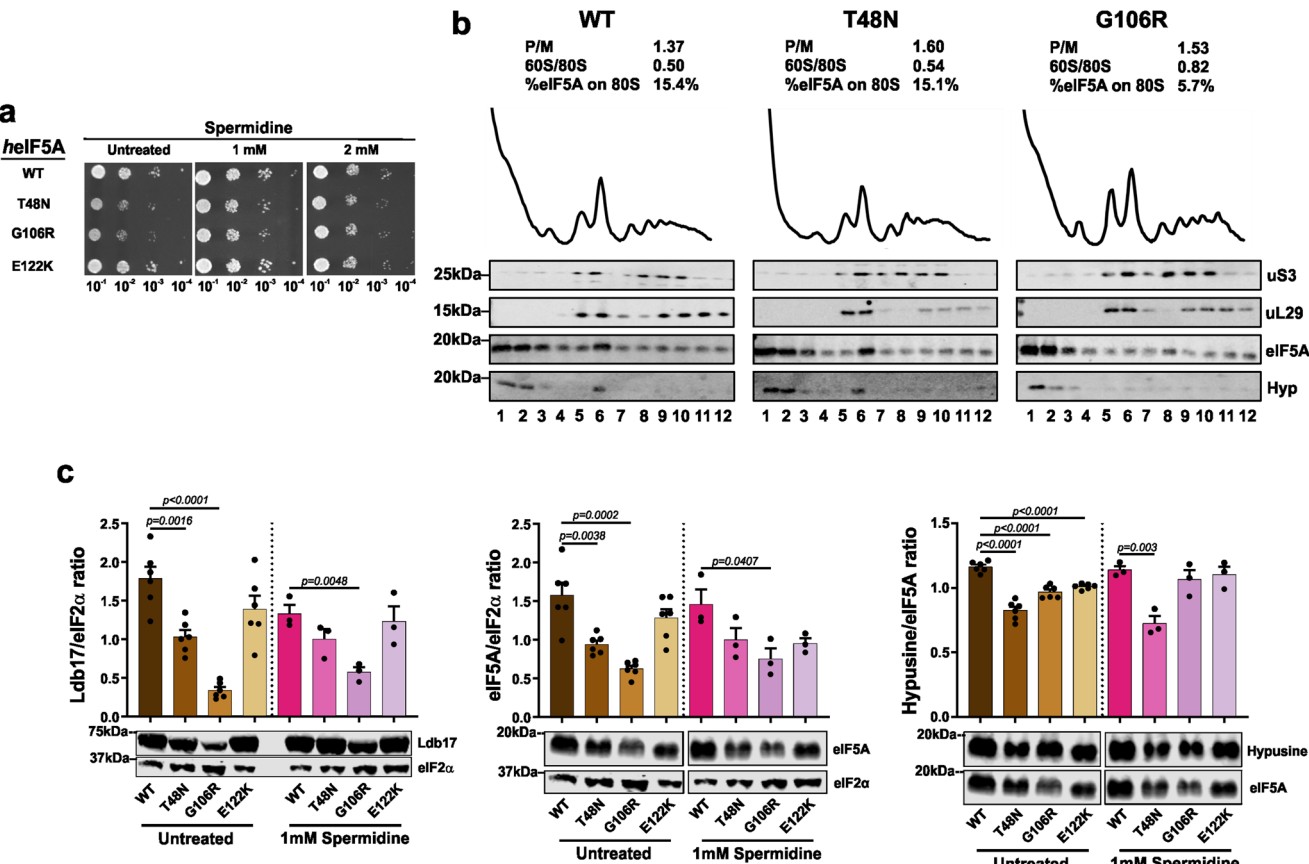

**Fig. 3 Spermidine partially rescues impaired eIF5A function phenotypes in yeast. a** Spermidine rescues growth of p.T48N and p.G106R strains. Growth of yeast strains for the human eIF5A (*he*IF5A) and the missense variants in minimum media supplemented with 0 (Untreated), 1 and 2 mM of spermidine at 30 °C. Eighteen replicates per temperature were performed. **b** Spermidine increases the interaction of *he*IF5A-T48N and *he*IF5A-G106R with ribosomes. Polysome profiles of *he*IF5A-WT, *he*IF5A-T48N and *he*IF5A-G106R grown at 30 °C in SC with 1 mM spermidine. Corresponding western blot analyses as described in the legend to Fig. 2c. Full uncropped images of gel blots are shown in Supplementary Fig. 12. One profile was performed. **c** Spermidine increases the synthesis of Ldb17 reporter in both p.T48N and p.G106R strains. Comparison of Ldb17 (left graph), total eIF5A (middle graph) and hypusinated eIF5A (right graph) among *he*IF5A yeast strains, grown at 30 °C in SCGal with and without 1 mM spermidine. Data presentation and statistics as described in the legend to Fig. 2b. Full uncropped images of gel blots are shown in Supplementary Fig. 13.

fixation. Supplementation with 1 µM spermidine resulted in partial rescue of micrognathia when compared to control MO (Fig. 4b, c, right graph), demonstrating that spermidine supplementation can rescue this developmental defect by bypassing loss of eIF5A. Thus, endogenous concentrations of spermidine within the yolk are not sufficient to overcome loss of eIF5A and spermidine supplementation is likely able to promote peptide synthesis to rescue this developmental defect[33].

## Discussion

*EIF5A* is a unique and critical gene for synthesis of proteins, especially those with PPTs. It is highly intolerant to variation, but so far no human condition caused by variants in this gene has been identified. We define a previously undescribed human disorder caused by heterozygous *EIF5A* variants. This is supported by high constraint for deleterious *EIF5A* variants in population databases, the de novo nature of all the variants described here, their absence from population databases along with high evolutionary conservation and the phenotypic similarity of patients ascertained via their genotypes (Fig. 1a–c). The disorder can be caused by protein truncating or missense variants. Of note, the codon encoding Arg109 was impacted in three out of seven cases. In humans, this amino acid is encoded by a CpG including codon (CGA) that may be prone to methylation, deamination and CG-TA transition, each of which could explain clustering of the

mutations seen in this study[38]. The phenotype of the condition consists of variable degrees of developmental delay, intellectual disability, microcephaly and craniofacial dysmorphism, including micrognathia (Table 1, Fig. 1a and Supplementary Note 1). Although both *EIF5A* and *EIF5A2* are hypusinated and widely expressed in adult human tissues, the expression of the former is ~20-fold higher in brain structures than the latter[16]. While *Eif5a*[gt/gt] mice are embryonically lethal[3], *Eif5a2*[−/−] mice are viable and display normal development[39]. Therefore, *EIF5A2* expression may not be sufficient to compensate the loss of *EIF5A* function[16]. Pathogenic variants in eIF2B subunits (*EIF2B1, EIF2B2, EIF2B3, EIF2B4* and *EIF2B5*; MIM #603896)[40], *EIF2S3* (MIM #300148)[41], *EIF3F* (MIM # 618295)[42], *EIF4E* (MIM #615091)[43–45] and *EIF4G1* (MIM #614251)[46] have been previously described to cause distinct neurological disorders. Our findings add to this list of translation factors implicated in human developmental disorders.

Yeast growth assays showed the deleterious nature of the truncating and p.T48N and p.G106R missense *EIF5A* variants (Fig. 2a). Our results with the truncating variant are concordant with a previous study that demonstrated that deletion of either eIF5A amino- or carboxy-termini were lethal in yeast[28]. For all studied missense variants, including the E122K variant, the polysome profiles were abnormal further indicating their deleterious nature. More specifically, the higher P/M ratios typically

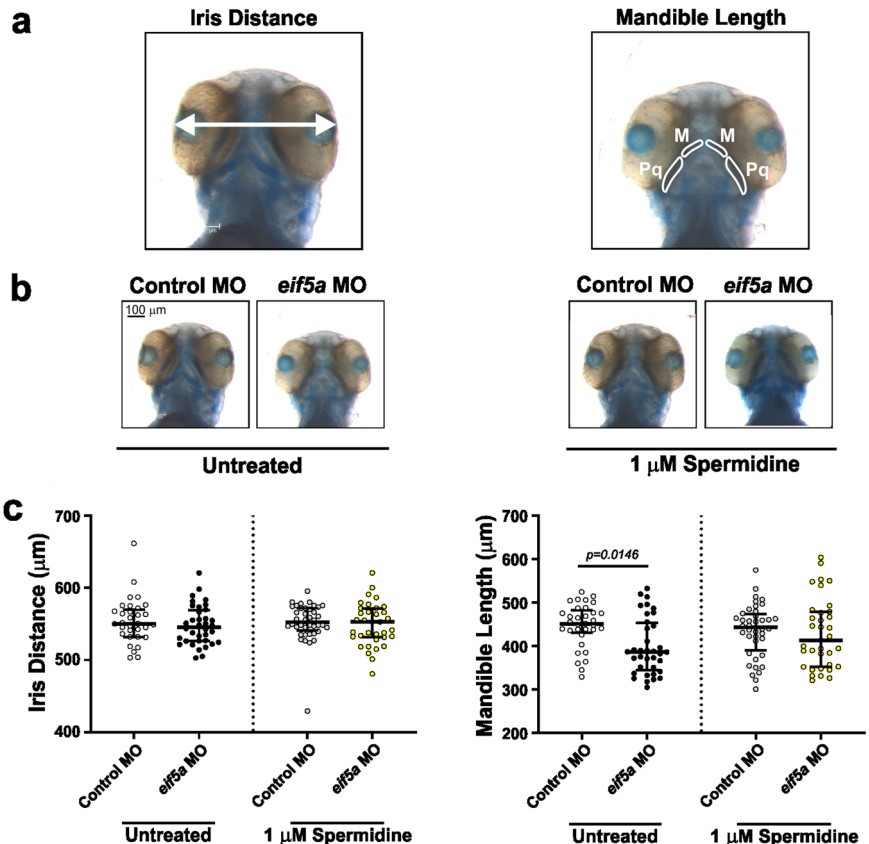

**Fig. 4 Knockdown of *eif5a* in zebrafish embryos induces micrognathia, which can be partially rescued by spermidine. a** Measurement of iris distance was used for assessing head size (left photograph), and both Meckel's (M) plus both palatoquadrate (Pq) cartilages for mandible assessment (right photograph). **b** Spermidine partially rescues micrognathia. Representative images of alcian blue-stained zebrafish larvae injected with 400 pg (1 nL) of either control or *eif5a* MOs. Larvae were incubated at 28 °C for 3 h in standard E3 embryo medium (SE3EM) and then transferred to either fresh SE3EM or fresh SE3EM plus 1 μM spermidine at 28 °C, and processed after 77 h post-fertilisation. **c** Quantification of measurements shown in panel **a**. Each data point corresponds to one fish, the longer horizontal line represents the median, whereas the shorter horizontal lines are the 25th and 75th percentiles. *P* values were determined by Kruskal Wallis test with multiple comparisons. *P* values for comparisons against control MO injected fish are provided when less than 0.05.

indicate a reduction in translation elongation rates or pausing of ribosomes causing 'traffic jams' on individual mRNAs leading to increased ribosome accumulation on mRNAs. The increase in the free 60S peak heights and the 60S/80S ratios (Fig. 2c) suggests that *EIF5A* missense variants may result in delay of 60S joining during initiation, or reduce the stability of 80S complexes. Both defects are consistent with the known functions of eIF5A in stimulating the first peptide bond formation and later during elongation and demonstrate clearly that each eIF5A missense mutation impacts protein synthesis globally[2].

Absence of the mutant transcript with the c.324dupA variant and lower level of *EIF5A* mRNA levels in the LCLs of individual 3 suggest haploinsufficiency as the underlying mechanism for the protein truncating variant. This is supported by phenotypic overlap of intellectual disability, microcephaly and retrognathia in patients with 17p13.1 microdeletions that encompass *EIF5A*[47]. Notably, the minimum critical region of this 17p13.1 microdeletion syndrome includes 17 protein-coding genes and its main phenotypic driver has not been resolved. Our observations suggest that haploinsufficiency of *EIF5A* could be responsible for the phenotype of 17p13.1 microdeletion syndrome.

The loss of function of the *EIF5A* missense variants is likely to be due to different mechanisms. The observation of reduced hypusination in cells expressing the p.T48N variant suggests that this variant impairs hypusination of the adjacent K50 residue. Other missense mutations changing residues adjacent to K50 also

have reduced hypusination[28], consistent with the idea that they impair interactions with DHPS and/or DOHH enzymes responsible for hypusination. This is also supported by our spermidine supplementation experiment (Fig. 3) that showed no improvement of p.T48N hypusination, indicating that spermidine levels are not a limiting factor for hypusination of this mutant. The DHPS molecular structure indicates that the surface surrounding its active site tunnel is highly acidic[48], while the eIF5A surface region around K50 has a complementary positive charged. In silico modelling (Fig. 1d) of the variant structure suggests T48N modestly increases the positive charge, but if or how this affects DHPS or DOHH interactions with eIF5A will require further experimentation. Reduction of total eIF5A levels in mutant cells (Figs. 2b and 3c) could be indicative of reduced protein or mRNA stability, but reduced protein levels per se are unlikely sufficient to explain the mutant phenotype. Instead our polysomal profile results are indicative of reduced interaction of eIF5A with ribosomes (Fig. 2c) as the common mechanism in the yeast model. As we did not have access to patient cells we could not test mRNA and protein expression levels or functional changes in individuals with the missense variants. Therefore, we cannot rule out that other mechanisms such as abnormal splicing or mRNA degradation may contribute to the disorder in individuals with missense *EIF5A* variants. Identification of more individuals in the future with disease-causing variants in this gene will be of great interest to uncover the underlying mechanisms.

We demonstrated that the loss of eIF5A function resulting from the variants has a deleterious impact on synthesis of proteins with PPT (Fig. 2d). The reduction was statistically significant for p.T48N and p.G106R variants. Although we did observe a reduction for the p.E122K variant, it did not reach statistical significance. In all our assays p.E122K was the least affected mutant. This is consistent with the milder individual phenotype of the patient harbouring this variant (Table 1 and Fig. 1a). Human KMT2D and SF3B4 are two MAGs that encode proteins with one of the highest number of PPTs (Fig. 2e). Interestingly, the phenotypes resulting from their loss of function resemble that of the individuals with EIF5A variants. From these data we surmise that impaired synthesis of proteins enriched in PPTs such as KMT2D and SF3B4 may underlie the phenotype(s) caused by defective eIF5A. Genes encoding the highest level of polyprolines display a strong association with biological processes such as actin/cytoskeletal-associated functions, RNA splicing/turnover, DNA binding/transcription and cell signalling[6]. Notably, variants in actin-encoding genes are known to cause human developmental disorders with microcephaly[49–51]. Similarly mandibular and craniofacial features of spliceosomal disorders overlap with patients described here[52]. Other possible lines of investigation could also be explored. For example, eIF5A regulates pancreatic cancer metastasis by modulating expression of RhoA and ROCK[53]. Germline variants in several Rho-GTPases cause developmental disorders with microcephaly[54,55]. Recently it was shown that hypusinated eIF5A promotes the efficient expression of a subset of mitochondrial proteins involved in the TCA cycle and oxidative phosphorylation[56]. Exploring the effects of these mutations on synthesis of mitochondrial proteins and function could be an interesting avenue. Additionally, eIF5A has functions in translation termination[10,11] that we did not explore in this study but which may contribute to this developmental disorder. The importance of eIF5A in neurodevelopment is further emphasised by the recent identification of DHPS deficiency in patients with a neurodevelopmental disorder with seizures and speech and walking impairment (MIM # 618480)[57]. The role of eIF5A in normal brain and craniofacial development has never been examined before to the best of our knowledge and the mechanism of how impaired eIF5A results in abnormal neurodevelopment will need to be studied in the future.

We demonstrated partial rescue of impaired eIF5A function and its resultant phenotypes in yeast and zebrafish models by spermidine (Figs. 3a–c and 4c–d). In yeast, spermidine rescued ribosome association defect and improved the polysome profile and PPT synthesis of the tested mutants. These effects are all consistent with rescuing eIF5A function. However, the molecular mechanism that underlies the rescue remains unclear. As eIF5A expression and hypusination was not increased by spermidine supplement (Fig. 3c), it appears spermidine is not acting via hypusination alone. In agreement with this idea, eIF5A-80S ribosome interaction was enhanced by spermidine (Fig. 3b). How spermidine could boost remaining eIF5A function is not clear. As spermidine rescued transient knockdown of eif5a in zebrafish, it may act similarly to the yeast model, or independently of eIF5A. Of note, our zebrafish model was based on transient knockdown of eif5a, and stable germline mutants will need to be studied to resolve these questions in the future. Importantly, spermidine supplementation has been shown to be safe and well tolerated in mice and humans[58]. It promotes longevity in yeast[35] and may extend the lifespan of mice and humans[59,60]. Higher spermidine intake has been shown to be linked to lower mortality in humans[61]. It has also been shown to protect against α-synuclein neurotoxicity in fruit flies[62] and its levels are important for memory-retrieval[63] and age-related memory-associated brain structures in rodents[64]. There is a growing interest in using spermidine as a therapeutic agent in conditions such as cognitive decline[65]. Although the effects of spermidine may be independent of its role in the synthesis of hypusinated eIF5A, our results raise the interesting possibility of a potential future therapy for individuals with EIF5A variant-associated disease.

In summary, we have defined a potentially treatable previously undescribed human Mendelian disorder caused by EIF5A mutations that result in reduced eIF5A–ribosome interactions via mutation-specific mechanisms. The phenotypes are likely explained by impaired synthesis of specific PPT-rich proteins. These findings uncover the role of eIF5A, and proteins with PPTs, in human brain and craniofacial development. Our findings open the avenue for future studies to identify the specific 'hard to synthesise' proteins, and the biological processes, most dependant on eIF5A function.

## Methods

**Ascertainment and exome sequencing analyses.** Seven individuals from seven unrelated families with heterozygous, de novo, variants in EIF5A were included in this study after informed consents were obtained. The Central Manchester, Cambridge South, and the Republic of Ireland RECs approved this study (02/CM/238, 10/H0305/83 and GEN/284/12, respectively). Informed consent for research studies from patients or their legal representatives was obtained in all cases. The authors affirm that human research participants provided informed consent for publication of the images in Fig. 1. The patients were followed up by clinical geneticists from France, the United Kingdom, and the USA. Variants were identified by trio whole-exome sequencing for the detection of an undiagnosed neurodevelopmental disorder associated with multiple congenital anomalies, following published methodology for sample and library preparation, sequencing data production, analysis and interpretation[66–69]. For the interpretation process we also considered the impact of variants on the preferentially expressed transcript according to the GTEx project[16] and the canonical protein according to UniProtKB[18], and the tolerance of the gene to variation according to gnomAD[19] scores, considering only high-quality non-flagged variants present in the 'only controls' subset of this database. We considered variants in constrained coding regions only (percentile > 90) according to Havrilla, et al.[23]. The clinical observations were gathered through the Matchmaker Exchange[21] and GeneMatcher[22] initiatives.

DNA samples from affected individual 3 and her parents, as well as informed consent, were obtained for PCR and Sanger sequencing confirmation. PCR was performed using primers 5′-AATGGCAGGAGAGGGTGTTT-3′ and 5′-TGCAGGTTCAGAGGATCACT-3′ and the GoTaq® Hot Start Green Master Mix 2x (PROMEGA). PCR products were purified using an AxyPrep™ Mag PCR Clean-Up kit (Axygen) and sequenced with a BigDye® Terminator v3.1 Cycle Sequencing kit (Applied Biosystems) on an ABI 3730xl DNA sequencer (Applied Biosystems). The resulting ABI files were examined using the Genome Assembly Program version 4.8b1. Similar Sanger sequencing methods were used to confirm the presence of the EIF5A variant in individual 1, and its absence in her parents.

**In silico analysis of variants and MAGs.** The evolutionary conservation analysis of the residues affected by missense variants was performed using the eIF5A canonical protein sequence from Homo sapiens (UniProtKB entry P63241), Mus musculus (UniProtKB entry P63242), Gallus gallus (UniProtKB entry Q09121), Danio rerio (UniProtKB entry Q6NX89, zebrafish hereafter), Drosophila melanogaster (UniProtKB entry Q9GU68) and Saccharomyces cerevisiae (UniProtKB entry P23301, yeast hereafter) using the ClustalWS alignment in Jalview[70] Version 2. 2.10.5. These residues were modelled in the yeast 60S ribosomal subunit with A-site tRNA, P-site tRNA and eIF5A (PDB entry 5GAK) using Chimera[71] 1.12.

We searched for all genes associated with microcephaly (MAG) deposited in OMIM until 6 February 2018 using the search criteria 'microcephaly (Entries with: gene map locus; Prefixes: +, #; Retrieve: gene map)'. Data were then merged between the OMIM search and the proline content of the human proteome analysed by Morgan and Rubenstein[30] and depicted in Supplementary Data 1.

**Gene, variants, morpholinos and plasmid synthesis and expression.** A yeast expression plasmid for human eIF5A (heIF5A-WT) was designed to express the human canonical protein sequence, but using the yeast optimised codon usage and placed in the context of the yeast TIF51A (yeIF5A) 5′ and 3′ regions. This was commercially synthesised (Epoch Life Sciences) and cloned into a pUC19 vector (Supplementary Note 2), resulting in plasmid pAV2578 (Supplementary Table 1). The gene was subsequently excised using XhoI and SpeI and cloned into SalI-and-SpeI digested single-copy-number (sc) LEU2 (YCplac111), URA3 (YCplac33) and high-copy-number (hc) LEU2 YEplac181 vectors, generating plasmids pAV2580, pAV2592 and pAV2593 (Supplementary Table 1). The heIF5A sequence was verified by Sanger sequencing at Eurofins Genomics using the M13 reverse primer.

The variants p.T48N, p.G106R, p.R109Tfs*8 and p.E122K, detected in individuals 1–3 and 7, respectively, were created through site-directed mutagenesis using primers (Supplementary Table 2), the sc pAV2580 plasmid, and the QuikChange Site-Directed Mutagenesis Kit (Agilent Technologies), following the manufacturer's instructions. An hc version of p.R109Tfs*8 was created in pAV2593. The resulting plasmids pAV2584[heIF5A-T48N], pAV2585[heIF5A-G106R], sc pAV2586[heIF5A-R109Tfs*8], pAV2587[heIF5A-E122K] and hc pAV2590[heIF5A-R109Tfs*8 LEU2] (Supplementary Table 1) were verified by Sanger sequencing as above.

For genetic knockdown of the eif5a gene in zebrafish, an eif5a splice site morpholino (MO) was synthesised (Gene Tools, Philomath, OR) for inhibition of the eif5a gene (5′-AACCCTATCCAAACATTACCTTTGC-3′) as previously published[36]. A standard control MO (5′-CCTCTTACCTCAGTTACAATTTATA-3′) by Gene Tools was also used.

**Leucocyte transformation and harvesting**. Peripheral blood mononuclear cells from one healthy, adult female (Control 1), one healthy, adult male (Control 2), and Individual 3 were transformed by Epstein-Barr virus into LCLs following a published protocol[72]. While five million LCLs from Individual 3 were harvested from five cell culture flasks under standard conditions, another five million LCLs from the same individual and number of flasks were treated for 6 h with 200 µg/mL of puromycin before harvesting, and both type of samples were kept at −80 °C before RNA extraction.

**Yeast strain construction and growth assays**. The J696 haploid yeast strain deleted for both yeast eIF5A (yeIF5A) genes and whose growth is supported by a plasmid bearing yeIF5A (Supplementary Table 3), as well as the yeIF5A plasmids C3287 (pAV2569), C3288 (pAV2571) and C3294 (pAV2565), and hc PPT reporter plasmids C4351 (pAV2566) and C4353 (pAV2570) (Supplementary Table 1) were kindly provided by Thomas E. Dever, Laboratory of Gene Regulation and Development, National Institute of Child Health and Human Development, National Institutes of Health, Bethesda, USA[2,5].

To study the effect of heIF5A-WT and the variants in yeast, a series of yeast strains was created by transformation and plasmid shuffling of the plasmids described in Supplementary Table 1 into strain J696 to generate the strains described in Supplementary Table 3.

To analyse the effect on yeast growth, heIF5A-WT and its variants, strains GP7439 and GP7443 diploid for yeIF5A gene, and hybrid strains, GP7441, GP7447, GP7448, GP7449 and GP7450 containing one yeIF5A copy and one heIF5A copy (or yeIF5A control) were patched on minimal SD + tryptophan medium, and replica-printed to SD+tryptophan+uracil+5-fluoro-orotic acid plates to select for loss of the WT yeIF5A and create haploid strains GP7440, GP7446, GP7444, GP7455, GP7474 and GP7456 each bearing a different heIF5A or yeIF5A plasmid as the sole source of eIF5A. Strains were 10-fold serially diluted and spotted on appropriate selective media ± spermidine at indicated concentrations to record growth phenotypes.

**Protein extraction and WB**. To study the effect of variants on humanised eIF5A synthesis and hypusination, strains GP7444, GP7455, GP7456 and GP7474 were grown in synthetic complete minus leucine (SC−LEU) medium, and strains GP7469, GP7482, GP7484 and GP7485 were grown in synthetic complete minus leucine and uracil (SC−LEU−URA) medium to mid-log phase at 30 °C. To study the effect of variants on synthesis of HA-tagged Ldb17 and Eap1 reporters, all strains between GP7490 and GP7493 and between GP7500 and GP7503 were grown in SC drop-out medium containing dextrose (0.4%) and galactose (2%) (SCGal-LEU-URA) for 24 h at 30 °C to induce expression of the PPT reporters. Spermidine (1 mM) was added to media where indicated.

Ten OD$_{600}$ units of cells of each strain were harvested, washed and resuspended with 10% trichloroacetic acid, and broken with acid-washed glass beads (Sigma-Aldrich) in a bead-beater (Biospec Products) twice for 45 s. These cell extracts were then centrifuged at 20,000g and the pellets resuspended in acetone twice. All these procedures were carried out at 4 °C. After a final centrifugation, pellets were dried 10 min at 37 °C and solubilised in a protease inhibitor buffer (100 mM Tris-HCl [pH 8.0], 1% SDS, 1 mM EDTA, 1 mM PMSF and 1 cOmplete™ protease inhibitor tablet [Roche]) during 1 h at 37 °C. Then, NuPAGE® LDS Sample Buffer 4× (Invitrogen) and β-mercaptoethanol were added to the extracts, which were subsequently boiled at 95 °C during 5 min and cooled immediately. After a final spin at 4 °C, supernatants were harvested and electrophoresis was carried out using 9–12 µL of protein extracts, NuPAGE™ 4–12% Bis-Tris gels (Invitrogen) and NuPAGE™ MOPS SDS 20× running buffer (Invitrogen) during 50 min at 200 V. The Precision Plus Protein™ All Blue Prestained Protein Standard (Bio-Rad Laboratories Ltd, UK) was used as a molecular weight marker. Gels were blotted onto Amersham™ Protan™ 0.45 µm nitrocellulose membranes (General Electric Healthcare), and using Whatman™ 3 mm Chr chromatography paper (General Electric Healthcare), NuPAGE™ 20× transfer buffer (Invitrogen), and XCell™ Blot module (Invitrogen) during 1 h at 30 V. After blocking non-specific binding, the membranes were incubated overnight at 4 °C with specific mouse anti-eIF5A (1:10,000; BD Biosciences, #611977), rabbit anti-hypusine (1:1,000; EMD Millipore,

#ABS1064), mouse anti-Pab1 (1:5000; EnCor Biotechnology, #MCA-1G1), mouse anti-HA.11 (1:4000; BioLegend, #901513) or chicken anti-eIF2α (1:500; Cambridge Research Biochemicals, custom designed). Then, the membranes were incubated with a corresponding secondary fluorescent labelled donkey anti-chicken (P/N: 926-32218), and goat anti-rabbit (P/N: 926-32211) or anti-mouse antibodies (P/N: 926-32350) (IRDye® 800CW; LI-COR Biosciences) and the signal was developed using a LI-COR Odyssey® CLx Imaging System (LI-COR Biosciences) using default parameters. The area of every band was selected to fit it to the best curve of fluorescence.

The precise epitope recognised by the commercial eIF5A monoclonal antibody is not known. It was raised to a protein 58–154 and our work shows it cross-reacts with the R109Tfs*8 mutant (Supplementary Fig. 4), implying its epitope is between 58 and 108. As the antibody cross-reacts with all mutant forms tested here we assume that G106 is not part of its binding site on eIF5A. Pab1 encoding the yeast polyA-binding protein was used as a loading control for most assays. However, Ldb17 with HA tag and Pab1 co-migrate in SDS-PAGE gels and some residual fluorescence remained for both proteins after stripping. Therefore, eIF2α antibodies were used as a loading control for blots for experiments using Ldb17-HA-expressing strains. Full scans of representative western blots shown in Figs. 2b, d and 3c are depicted in Supplementary Figs. 9, 11 and 13.

**Polysome profile analysis**. The GP7444, GP7455, GP7456 and GP7474 strains were grown to mid-log phase at 30 °C in 80 ml of SC-LEU medium ± spermidine (1 mM). Fifty millimolar formaldehyde was added followed immediately by 15 ml of pre-made frozen media droplets. We used formaldehyde treatment to stabilise polysomes and bound factors rather than cycloheximide, because the latter has been found to enhance eIF5A–ribosome interactions[24]. Cross-linking was then carried out on ice water for 1 h before quenching with glycine (final concentration of 100 mM). Whole-cell extracts were prepared by bead beating in lysis buffer (20 mM HEPES, 2 mM magnesium acetate, 100 mM potassium acetate, 0.5 mM DTT), clarified (10,000g 10 min) and separated on 15–50% sucrose gradients by centrifugation at 40,000 r.p.m. for 2.5 h using a SW41 Beckman rotor. Gradients were fractionated while scanning at $A_{254}$ to visualise the indicated ribosomal species. Polysome-to-monosome (P/M) and 60S/80S ratios were calculated by comparing the areas under the 60S, 80S and polysome peaks. Gradient fractions were TCA precipitated and underwent western blot analysis using antibodies as described above as well as Rps3(uS3) (1:50,000) and Rpl35(uL29) (1:10,000) rabbit polyclonal antibodies (a kind gift from Dr Martin Pool, University of Manchester). Full scans of representative western blots shown in Figs. 2c and 3b are depicted in Supplementary Figs. 10 and 12, respectively.

**RNA extraction and quantitative real-time PCR (qRT-PCT)**. Total RNA was extracted using RNeasy Mini kit (Qiagen) according to the manufacturer's protocol from one million LCLs of Control 1 and Control 2 each, and from one million, untreated LCLs and one million puromycin-treated LCLs of Individual 3. RNA concentration was measured using a NanoDrop 2000 spectrophotometer (Thermo Scientific). RNA was reverse transcribed with random hexamer primers (Promega) to generate cDNA using the M-MLV Reverse Transcriptase kit (Promega) according to the manufacturer's protocol. qRT-PCR reactions were performed on a Bio-Rad CFX394 Real Time system (Bio-Rad) using Power SYBR Green PCR Master mix (Applied Biosystems) and the forward 5′-GCCATGTAA-GATCGTCGAGA-3′ and reverse 5′-GGAGCAGTGATAGGTACCCA-3′ EIF5A primers (Sigma-Aldrich). The level of EIF5A mRNA was evaluated using a relative quantification approach (2−ΔCT method) with human GAPDH and its primers 5′-ATGGGGAAGGTGAAGGTCG-3′ and 5′-TAAAAGCAGCCCTGGTGACC-3′ as the internal reference. To detect if the frameshift-encoding transcript is expressed, we performed bidirectional Sanger sequencing of Individual 3's EIF5A cDNA, obtained from both untreated and puromycin-treated LCLs, using the aforementioned forward primer and reverse 5′-GCCTTGATTGCAACAGCTGC-3′ primer, as previously described.

**Zebrafish knockdown, staining and imaging**. Zebrafish husbandry was approved by The University of Manchester Ethical Review Board and all experiments were performed in accordance with UK Home Office regulations (PPL P132EB6D7). Four hundred picograms (in 1 nL) of either the eif5a or control MOs were injected into 25–75 fertilised nacre embryos per condition, at the single-cell stage. Injected embryos were incubated at 28 °C in standard E3 embryo medium (SE3EM) until 3 hpf and then split into two groups, one of them only in SE3EM and the other one in SE3EM plus Spermidine (85558; Sigma-Aldrich) to a final concentration of 1 µM. Both groups were incubated at 28 °C until 77 hpf, without renewing the media.

At 77 hpf, larvae were terminated using 4% MS222 and fixed for 1 h with 2% paraformaldehyde, and then stained using a two-colour acid-free bone and cartilage staining protocol[73]. Stained larvae groups were blinded to a manipulator, mounted into 4% methylcellulose and imaged using a DFC7000 T Camera (Leica) coupled to a M165 FC Microscope (Leica) using ×10 objective and LASX image capture software (version 3.4.2; Leica). Head size and mandibles were measured as depicted in Fig. 4a using ImageJ software (version 1.52j).

**Statistics**. All statistics were calculated with either GraphPad Prism 8.3.0 (GraphPad Software) or SPSS v25 (IBM) and all experiments were carried out using three biological (yeast and zebrafish analyses) or technical (Individual 3's *EIF5A* RNA expression) replicates. Chi-square was performed to study the association between content of PPTs and MAGs. Signals of WBs from yeast extracts were normalised to the mean signal across each blot for each antibody and for loading variance between lanes using similarly normalised control antibody signals as follows in Eq. (1):

$$ab\ ratio = \frac{Tab}{Mean\ Tab} \bigg/ \frac{Cab}{Mean\ Cab} = \frac{Tab \times Mean\ Cab}{Mean\ Tab \times Cab}, \tag{1}$$

where Tab is the test antibody and Cab the control antibody. For hypusine quantification, eIF5A was used as its control antibody. Unpaired *t*-test against control samples was performed for western blots from yeast extracts and for Individual 3 *EIF5A* RNA expression. These results were depicted using bar plots, which represent the mean (average) plus standard error of the mean with overlaid data points representing independent experiments. Kruskal Wallis test with multiple comparisons was performed for zebrafish analyses. These results were depicted using dot plots and the longer horizontal line represents the median, whereas the shorter horizontal lines are the 25th and 75th percentiles. A *P* value below 0.05 was considered significant, which is given at the top of each graph, where relevant.

**Reporting summary**. Further information on research design is available in the Nature Research Reporting Summary linked to this article.

## Data availability

All relevant data supporting the key findings of this study are available within the article and its Supplementary Information files or from the corresponding author upon reasonable request. Source data are provided with this paper.

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

## Acknowledgements

We are thankful to all the individuals and their families for taking part in the study. We thank Tom Dever (National Institutes of Health, USA) and Beth Grayhack (University of Rochester Medical Center, USA) for kind gifts of plasmids and yeast strains used in this study, as well as Martin Pool (University of Manchester, UK) for the gift of antibodies to ribosomal proteins. We are thankful to the Deciphering Developmental Disorders (DDD) study for the invaluable collaboration. The DDD Study (Cambridge South REC approval 10/H0305/83 and the Republic of Ireland REC GEN/284/12) presents independent research commissioned by the Health Innovation Challenge Fund (grant number HICF-1009-003), a parallel funding partnership between the Wellcome Trust and the Department of Health and the Wellcome Trust Sanger Institute (grant number WT098051). The views expressed in this publication are those of the author(s) and not necessarily those of the Wellcome Trust, BBSRC or the Department of Health. The research team acknowledges the support of the National Institute for Health Research, through the Comprehensive Clinical Research Network, UK. V.F. acknowledges to CONICYT, Chile's National Commission for Scientific and Technological Research, for its scholarship support (grant number 72160007). V.F., W.G.N. and S.B. acknowledge to the Kabuki Research Fund at Manchester University NHS Foundation Trust. W.G.N. acknowledges support from Action Medical Research (GN2494), and the Manchester NIHR Biomedical Research Centre (IS-BRC-1215-20007). G.D.P. and M.D.J. acknowledge to Biotechnology and Biological Sciences Research Council (BBSRC), UK, for its financial support (grant BB/N014049/1). P.R.K and S. Crilly were supported by the Stroke Association (TSA LECT 2017/02) and the NC3Rs (NC/N002598/1). J.A. and C.T.G. were supported by the Agence Nationale de la Recherche (CranioRespiro project and 'Investissements d'avenir' program (ANR-10-IAHU-01)) and MSDAvenir (Devo-Decode project).

## Author contributions

S.B. conceived the project. S.B., G.D.P., P.R.K, W.G.N., V.F. and M.D.J. designed the study. S.J.D., A.G.L.D., A.E.F., V.H., J.A., D.L., P.N., J.R., M.S., L.A.C., C.J.S., B.R.S., J.L.G. and C.T.G provided patients' phenotypes and collected samples. V.F. performed compilation of patients' clinical data, in silico analysis of variants, association of PPT with MAGs, Sanger sequencing confirmation, synthesis of gene, variants and yeast plasmids, cell transformations, yeast growth assays and western blots. M.D.J. performed polysome fractionation and western blot analysis. S.L., S. Crilly and S.E.W. performed zebrafish injections, staining and imaging. S. Cuvertino performed RNA extraction, expression analysis and sequencing of cDNA, and western blot of human samples. V.F., S.B., G.D.P., P.R.K. and M.D.J. wrote the manuscript. All authors reviewed, edited and approved the final manuscript.

## Competing interests

The authors declare no competing interests.

## Additional information

[1]Division of Evolution & Genomic Sciences, School of Biological Sciences, Faculty of Biology, Medicine and Health, University of Manchester, Manchester, UK. [2]Laboratorio de Genética y Enfermedades Metabólicas, Instituto de Nutrición y Tecnología de los Alimentos (INTA), Universidad de Chile, Santiago, Chile. [3]Division of Molecular and Cellular Function, School of Biological Sciences, Faculty of Biology, Medicine and Health, University of Manchester, Manchester, UK. [4]Manchester Academic Health Science Centre, University of Manchester, Manchester, UK. [5]Division of Neuroscience & Experimental Psychology, School of Biological Sciences, Faculty of Biology, Medicine and Health, University of Manchester, Manchester, UK. [6]Institute of Medical Genetics, University Hospital of Wales, Cardiff, UK. [7]Wessex Clinical Genetics Service, Princess Anne Hospital, Southampton, UK. [8]Human Development and Health, Faculty of Medicine, University of Southampton, Southampton General Hospital, Southampton, UK. [9]Division of Cancer and Genetics, School of Medicine, Cardiff University, Cardiff, UK. [10]Department of Genetics, AP-HP, Hôpital Necker Enfants Malades, Paris, France. [11]1Laboratory of Embryology and Genetics of Human Malformations, INSERM UMR 1163, Institut Imagine, Paris, France. [12]Paris Descartes-Sorbonne Paris Cité University, Institut Imagine, Paris, France. [13]Manchester Centre for Genomic Medicine, St Mary's Hospital, Manchester University NHS Foundation Trust, Health Innovation Manchester, Manchester, UK. [14]Division of Genetics and Metabolism, Department of Pediatrics, University of South Florida, Tampa, FL, UK. [15]Northern Genetics Service, Institute of Genetic Medicine, Newcastle upon Tyne, UK. [16]Division of Clinical Genetics, Children's Mercy, Kansas City, MO, USA. [17]Department of Pediatrics, University of Missour—Kansas City, Kansas City, MO, USA. [18]Center for Pediatric Genomic Medicine Children's Mercy, Kansas City, MO, USA. [19]School of Medicine, University of Missouri–Kansas City, Kansas City, MO, USA. [20]Department of Pathology and Laboratory Medicine, Children's Mercy, Kansas City, MO, USA. [21]Division of Genetics and Genomic Medicine, Department of Pediatrics, Washington University School of Medicine, St. Louis, MO, USA. ✉email: paul.kasher@manchester.ac.uk; graham.pavitt@manchester.ac.uk; siddharth.banka@manchester.ac.uk

