## [Peer Review File · Nature Communications]

REVIEWER COMMENTS

Reviewer #1 (Remarks to the Author):

Faundes and colleagues link mutations of EIF5A, a translation factor that is specifically post-translationally modified by the polyamine spermidine to Kabuki syndrome-like developmental defects. The authors further suggest that spermidine supplementation might improve symptoms associated to EIF5A deficiency. Overall the phenotyping data on EIF5A variant defects is solid. However, the reasoning for spermidine supplementation is not entirely clear because no mechanistic details are provided regarding the compensation of neither the hypusination defect nor hypusination-independent defects.

Specific points:

- It has been published recently, that eIF5A hypusination is causally linked to mitochondrial phenotypes like respiration. <https://www.ncbi.nlm.nih.gov/pubmed/31130465>
Therefore, it should be tested, if the eIF5A mutations introduced to model systems impair respiration and/or other mitochondrial phenotypes.
- In general, the quantifications of the western blots are difficult to follow based on the representative blots. Most bands look overexposed. Please (i) add a brief description of the quantification procedure in the methods section, (ii) full scans of the representative blots in a supplementary data file and (iii) molecular weight indicators for all blot images.
- Add an explanation, why different normalization procedures were used. Some protein levels are normalized to eIF2a, others to Pab1. Why?
- eIF5a monoclonal antibody: immunization peptide spans aa 58-154. Two of the mutations lie within this region. How can the authors ensure equivalent epitope-paratope affinities? Could this affect assessing the expression levels?
- The reasoning behind supplementing spermidine is not entirely clear. Do the authors assume that spermidine levels becomes rate-limiting for hypusination in haploid mutant eIF5A backgrounds? If that is the case they should measure spermidine levels under these experimental conditions to justify supplementation. As the authors suggest and the zebra fish data strongly implies, spermidine supplementation might bypass impaired eIF5A function. This does not explain, however, the improved translation of PPTs (see reference 32, which suggests that hypusinated eIF5A is critical for proline translation). Alternatively, do eIF5a mutant variants (in particular T48N) have differential binding affinities to deoxyhypusine synthase or deoxyhypusine hydroxylase, which are reversed back to the wild type situation upon spermidine supplementation?
It is critical to explore additional mechanisms by which spermidine supplementation could compensate for the eIF5A defects (e.g. general translation efficiency).
- In Figure 3, it is difficult to interpret the polysome profiles without having the untreated conditions alongside the spermidine-treated profiles.
- For all yeast experiments, the media pH must be carefully monitored as this is a major confounder of spermidine supplementation experiments and likely the reason for the growth defect in the high spermidine conditions.
- How was the spermidine concentration for the zebra fish experiments chosen?
- Study participants' (or their legal guardians') consent to publish non-anonymized pictures should be explicitly stated.

- Statistics: The authors state "Unpaired t-test with or without normalization against control samples and Welch's correction was performed for WBs from yeast extracts or Individual 3's EIF5A RNA expression, respectively." - This is not very informative. If a pre-normalization against control samples was applied resulting in a standard deviation=0 of the control samples, Welch's t-test (or any test that assumes normal distribution of data for that matter) is not very meaningful.

-The authors remain vague about the definition and delimitation of this novel syndrome: If it is a novel disorder, then the authors should label it with a name. Further: How exactly is the relation/demarcation to the Kabuki syndrome and other syndroms. This should be made much clearer in the introduction or/and the discussion. Also, it would be interesting to sequence EIF5A from more than one diagnosed Kabuki patients.

Reviewer #2 (Remarks to the Author):

The authors describe the identification of de novo mutations in the gene EIF5A1, define the clinical phenotype and the consequences on cellular translation by the aberrant encoded protein by several routes including polysome profiling. They use yeast and cellular systems to address their questions relate to this crucial and conservative protein.

The study is impressive, both from clinical point of view and the underlying biology that is thoroughly and convincingly studied. I have hardly comments other than the advice to publish this great work!
Few comments:

- I found the designation of 'a syndrome' in the title not appropriate as it implicates a specific condition. There are 7 cases with overlapping features described, but heterogeneous. My advise to use a term such as 'Mendelian disorder' than 'a syndrome'.

- authors may explain bit more the relation to the role of EIF5A2; they state that the protein is undetectable and therefore do not refer to this any further, but it seems to be significantly expressed in adult brain. Could this paralogue have any redundant effects/protection of the consequences of the loss of function of EIFA1?

- Th R109 is 3/7 times mutated in this cohort. Could authors provide any explanation why it seems that this position is mutated relatively frequently?

Reviewer #1 (Remarks to the Author)

1. It has been published recently, that eIF5A hypusination is causally linked to mitochondrial phenotypes like respiration. <https://www.ncbi.nlm.nih.gov/pubmed/31130465>. Therefore, it should be tested, if the eIF5A mutations introduced to model systems impair respiration and/or other mitochondrial phenotypes.

We thank the reviewer for drawing attention to the interesting recent findings where Puleston et al have shown that hypusinated eIF5A promotes the efficient expression of a subset of mitochondrial proteins involved in the TCA cycle and oxidative phosphorylation. Certainly, in light of these findings, mitochondrial proteins, respiration and/or other mitochondrial phenotypes will be excellent candidates to test although the phenotype in our patients is not typical for mitochondrial defects. We have added a sentence reflecting this (page 16).

The primary intentions of our work were to – (a) identify the novel disease; (b) perform initial delineation of the human phenotype; (c) Characterize the mechanism of how mutations caused loss of eIF5A function; (d) Explore the effects of loss of eIF5A function on mRNA translation; and (e) Assess if spermidine can rescue the phenotypes. Specifically, we have not ventured into identification of the specific proteins and the biological processes most affected by loss of eIF5A function. We believe that these questions will be best addressed via comprehensive studies in the future. Even if we wanted to explore these effects, our yeast system may not be the best model to test reliance on mitochondria. As yeast cells preferentially grow on glucose by fermentation to ethanol, they have a lower reliance on mitochondrial processes, especially when supplemented with all amino acids (which can otherwise be synthesized from TCA precursors). Although it is possible to grow cells on alternate non-fermentable carbon sources that rely on mitochondria, this significantly reduces the rate of growth of wild-type cells and growth rate is linked tightly to protein synthesis requirements. These confounding issues would make interpretation of the results challenging and may have limited applicability towards understanding of the pathology of the human disease. Taking these points into consideration, we assert that this line of investigation, while of clear academic interest, is beyond the scope of the current study.

2. In general, the quantifications of the western blots are difficult to follow based on the representative blots. Most bands look overexposed. Please (i) add a brief description of the quantification procedure in the methods section, (ii) full scans of the representative blots in a supplementary data file; (iii) molecular weight indicators for all blot images.

We are sorry that the reviewer found that the blot quantification was hard to follow. The western blots were developed with the Li-Cor system using fluorescent secondary antibodies rather than traditional film. This technique records information over a far broader range than a 256 grayscale monitor can discern. It is also generally acknowledged to be superior for quantification than many other methods.

In response to the specific critique and also the question raised concerning the use of statistics (point 10) we have now altered our quantification processes so that the errors can be seen across all samples, rather than normalising the WT to 1.

- (i) *We have included details of the quantification procedure in the expanded 'Statistics' section of the online methods (page 27).*
- (ii) *We have included new supplementary data figures S9 to S13 showing the full regions of the membranes captured and indicated on each the molecular weights and the region used in*

the cropped figures. In some cases blots were cut horizontally prior to incubation with primary antibodies so that a single membrane could be simultaneously probed with 2 antibodies. Also, the molecular weight marker used for WB is described in the methods section (page 23-25)

(iii) *Molecular weight markers have been added to each blot image shown in the main and supplementary figures.*

3. Add an explanation, why different normalization procedures were used. Some protein levels are normalized to eIF2a, others to Pab1. Why?

This was purely for technical reasons. Ldb17 with HA tag has a very similar MW to the Pab1 control we were using, and in initial experiments some residual fluorescence remained for both proteins after stripping. We therefore changed the control protein to avoid any potential issues distinguishing which signal is from which antibody. This is now clarified in the 'Protein extraction and Western Blotting (WB)' part of the methods section (page 24-25).

4. eIF5a monoclonal antibody: immunization peptide spans aa 58-154. Two of the mutations lie within this region. How can the authors ensure equivalent epitope-paratope affinities? Could this affect assessing the expression levels?

We have performed western blots with the frameshift variant R109fs. This reveals that this antibody recognises the truncated allele, implying that the monoclonal antibody targets an epitope before aa 109. We have added a New Supplementary figure (Figure S4) showing this new experiment which helps to clarify this point and provides additional information on the frameshift mutant's expression and hypusination in the yeast model system. We have described the data in the main manuscript to reflect this (page 8). Data suggests that E122K should be unaffected in its antibody recognition. This leaves only G106R within the possible epitope site, but the chance for this to impact our assays with G106R variant are minimal because monoclonal antibodies that function in western blotting typically recognise a single short linear epitope (generally no more than 6-9 residues) and a major amino acid side-chain change to arginine from a single hydrogen (glycine) would be expected to completely eliminate antibody-antigen binding at the stringency used in western blotting if G106 was within the epitope.*

Although we could not be sure at the outset of the study, clearly this antibody is able to recognise each of the mutants (see Fig. 2b). We did consider including an epitope tag (eg Myc/FLAG) on eIF5A so that we could be more sure of detecting expression levels, but were not able to find a reference where anyone had done this successfully in the past for eIF5A. We, therefore, could not be sure that any tag we introduced to such a small protein that interacts extensively with the ribosome would not adversely affect eIF5A function. So we decided to only use untagged eIF5A constructs. Importantly, for the ribosome interaction studies (Fig 2c) our assessment of the relative association (or not) with 80S subunits cannot be affected at all by any weakened antibody detection. Hence, our view is that none of the major conclusions should be affected by any antibody detection issues for the G106R or any other allele.

5. The reasoning behind supplementing spermidine is not entirely clear. Do the authors assume that spermidine levels becomes rate-limiting for hypusination in haploid mutant eIF5A backgrounds? If that is the case they should measure spermidine levels under these experimental conditions to justify supplementation.

We are sorry this reasoning was not clear. We have now added additional clarification for the thinking behind this approach (page 11). We now show hypusine quantified relative to eIF5A levels in Fig 3c. This makes it clearer that spermidine supplementation did not have any direct impact on hypusination of any of the eIF5A forms. Our data does not support that spermidine is rate-limiting here, so we have not measured polyamine levels.

As the authors suggest and the zebra fish data strongly implies, spermidine supplementation might bypass impaired eIF5A function. This does not explain, however, the improved translation of PPTs (see reference 32, which suggests that hypusinated eIF5A is critical for proline translation). Alternatively, do eIF5a mutant variants (in particular T48N) have differential binding affinities to deoxyhypusine synthase or deoxyhypusine hydroxylase, which are reversed back to the wild type situation upon spermidine supplementation?

It is critical to explore additional mechanisms by which spermidine supplementation could compensate for the eIF5A defects (e.g. general translation efficiency).

It appears that spermidine is likely having eIF5A dependent and independent actions that are separate from its role in hypusination and which together promote protein synthesis in general as well as promoting eIF5A-ribosome interactions and PPT synthesis. Resolving these issues will undoubtedly require significant additional study beyond the scope of this manuscript. We have expanded the discussion section to highlight these points (page 17).

6. In Figure 3, it is difficult to interpret the polysome profiles without having the untreated conditions alongside the spermidine-treated profiles.

We did not reproduce the untreated condition alongside to avoid repetition of data that is provided in Figure 2c. These experiments were done in parallel, but obviously take up a lot of space to display. We acknowledge that this does require a reader to check between figures. We hope that this will be easier in a final formatted pdf version. If the editor thinks it appropriate, we could prepare a supplementary figure that reproduces the data side-by-side.

7. For all yeast experiments, the media pH must be carefully monitored as this is a major confounder of spermidine supplementation experiments and likely the reason for the growth defect in the high spermidine conditions.

We did not monitor pH in our experiments. Although the possibility raised by the reviewer is valid, there is evidence in the literature showing that longevity induced by spermidine in yeast is pH-independent, even at higher concentrations of spermidine (4 mM) (<https://www.nature.com/articles/ncb1975>).

8. How was the spermidine concentration for the zebra fish experiments chosen?

We explored 10-fold dilutions, from 1 mM (1000 μ M) to 0.1 μ M. 100 μ M or higher concentrations killed larvae before fixation, at 10 μ M larvae looked fine at fixation time, but to have a secure range, we chose 1 μ M. We have added a sentence in the manuscript to clarify this (page 12).

9. Study participants' (or their legal guardians') consent to publish non-anonymized pictures should be explicitly stated.

Done (page 19).

10. Statistics: The authors state "Unpaired t-test with or without normalization against control samples and Welch's correction was performed for WBs from yeast extracts or Individual 3's EIF5A RNA expression, respectively." - This is not very informative. If a pre-normalization against control samples was applied resulting in a standard deviation=0 of the control samples, Welch's t-test (or any test that assumes normal distribution of data for that matter) is not very meaningful. We thank the reviewer for pointing this out. We have now changed our normalisation procedure so that variation across all samples is taken into consideration and repeated the statistical analysis for all Western blots (Page 27).

11. The authors remain vague about the definition and delimitation of this novel syndrome: If it is a novel disorder, then the authors should label it with a name. Further: How exactly is the relation/demarcation to the Kabuki syndrome and other syndromes. This should be made much

clearer in the introduction or/and the discussion. Also, it would be interesting to sequence EIF5A from more than one diagnosed Kabuki patients.

It is current usual practice to present new genetic conditions without naming them in the first instance. Although some patients were suspected to have a condition similar to Kabuki syndrome, this was not the case in all patients. We think that our description in the current manuscript achieves the right balance of emphasizing the relationship without over-interpreting the genetic and clinical data. We agree that it would be interesting to identify more 'EIF5A' patients with or without suspicion of Kabuki syndrome. We expect that publication of this paper will prompt such studies in the future.

Reviewer #2 (Remarks to the Author):

The authors describe the identification of de novo mutations in the gene EIF5A1, define the clinical phenotype and the consequences on cellular translation by the aberrant encoded protein by several routes including polysome profiling. They use yeast and cellular systems to address their questions relate to this crucial and conservative protein.

The study is impressive, both from clinical point of view and the underlying biology that is thoroughly and convincingly studied. I have hardly comments other than the advice to publish this great work!
We thank the reviewer for the positive comments.

- 1. I found the designation of 'a syndrome' in the title not appropriate as it implicates a specific condition. There are 7 cases with overlapping features described, but heterogeneous. My advice to use a term such as 'Mendelian disorder' than 'a syndrome'.**

We agree with the reviewer. We have replaced 'syndrome' with 'Mendelian disorder' in the title.

- 2. Authors may explain bit more the relation to the role of EIF5A2; they state that the protein is undetectable and therefore do not refer to this any further, but it seems to be significantly expressed in adult brain. Could this paralogue have any redundant effects/protection of the consequences of the loss of function of EIF5A1?**

This is an interesting idea. Of note, the expression of EIF5A in adult brain structures is 21x higher (~84 TPM) than EIF5A2 (~4 TPM) (GTEx project). We have now expanded our discussion to make reference to this (page 13). However, if or how the expression of EIF5A2 changes when EIF5A function is impaired remains to be explored.

- 3. The R109 is 3/7 times mutated in this cohort. Could authors provide any explanation why it seems that this position is mutated relatively frequently?**

We thank the reviewer for pointing this out. We had also noticed this 'clustering' in our data. In human EIF5A, Arg109 is encoded by a CpG including codon (CGA) and may be prone to methylation, deamination and/or CG-TA transition. We have expanded our discussion to make this point (page 13).

REVIEWERS' COMMENTS

Reviewer #1 (Remarks to the Author):

Good revision